# Clinical Efficacy of Sitafloxacin–Colistin–Meropenem and Colistin–Meropenem in Patients with Carbapenem-Resistant and Multidrug-Resistant *Acinetobacter baumannii* Hospital-Acquired Pneumonia (HAP)/Ventilator-Associated Pneumonia (VAP) in One Super-Tertiary Hospital in Bangkok, Thailand: A Randomized Controlled Trial

**DOI:** 10.3390/antibiotics13020137

**Published:** 2024-01-30

**Authors:** Manasawee Wantanatavatod, Panuwat Wongkulab

**Affiliations:** Division of Infectious Disease, Department of Medicine, Rajavithi Hospital, Bangkok 10400, Thailand; wongpanuwat@gmail.com

**Keywords:** *Acinetobacter baumannii*, HAP/VAP, carbapenem-resistant, sitafloxacin

## Abstract

Background: Carbapenem-resistant *A. baumannii* (CRAB) hospital-acquired pneumonia (HAP)/ventilator-associated pneumonia (VAP) is now a therapeutic problem worldwide. Method: An open-label, randomized, superiority, single-blind trial was conducted in Rajavithi Hospital, a super-tertiary care facility in Bangkok, Thailand. CRAB HAP/VAP patients were randomly assigned to receive either sitafloxacin–colistin–meropenem or colistin–meropenem. Outcomes in the two groups were then assessed with respect to mortality, clinical response, and adverse effects. Result: Between April 2021 and April 2022, 77 patients were treated with combinations of either sitafloxacin plus colistin plus meropenem (*n* = 40) or colistin plus meropenem (*n* = 37). There were no significant differences between the two groups with respect to all-cause mortality rates at 7 days and 14 days (respectively, 7.5% vs. 2.7%; *p* = 0.616, and 10% vs. 10%; *p* = 1). Patients who received sitafloxacin–colistin–meropenem showed improved clinical response compared with patients who received colistin–meropenem in terms of both intention-to-treat (87.5% vs. 62.2%; *p* = 0.016) and per-protocol analysis (87.2% vs. 67.7%; *p* = 0.049). There were no significant differences between the two groups with respect to adverse effects. Conclusions: Adding sitafloxacin as a third agent to meropenem plus colistin could improve clinical outcomes in CRAB HAP/VAP with little or no impact on adverse effects. In short, sitafloxacin–meropenem–colistin could be another therapeutic option for combatting CRAB HAP/VAP.

## 1. Introduction

Hospital-acquired pneumonia (HAP) and ventilator-associated pneumonia (VAP) are the most prevalent nosocomial infections [1]. According to the National Antimicrobial Resistant Surveillance Center, Thailand (NARST), between January and December 2020, the *Acinetobacter calcoaceticus–baumannii* complex was the second-most common organism isolated from sputum cultures [2]. VAP is the most frequently reported healthcare-associated *A. baumannii* infection; it is implicated in 3–7% of cases, leading to increased mortality rates [3]. Among patients requiring mechanical ventilation for more than 5 days, the prevalence of *Acinetobacter* spp. increases dramatically, accounting for 26% of respiratory infections in one series [4].

According to NARST, resistance to colistin was about 2.9% in 2018; this was lower than the corresponding figure for carbapenem resistance. The *Acinetobacter calcoaceticus–baumannii* complex remained susceptible to colistin. The minimum inhibitory concentrations 50 (MIC_50_) and 90 (MIC_90_) for colistin were determined to be 2 µg/mL, which is higher than the previously reported MIC_90_ valve of 1.5 µg/mL in 2017 [5]. In addition, analysis of data from 2000 to 2020 showed that, during this period, resistance to imipenem increased from 14.2% to 81.3%, and from 7.6% to 72.3%, in outpatient and inpatient department settings, respectively [2].

In Rajavithi Hospital, in 2021, *A. baumannii* was the pathogen most commonly isolated from sputum (21.4%). Cumulative antibiogram data showed that the susceptibility of *A. baumannii* to meropenem was only 6% but was 95% to colistin. Hospital data concerning susceptibility to sitafloxacin were limited, however.

Before the release of the Infectious Diseases Society of America (IDSA) 2021 guidance, there was no clear standard-of-care antibiotic regimen for CRAB infections. The guidance enabled medical practitioners treating infections caused by CRAB to be better informed, and also helped to support research. Seven randomized controlled trials have investigated possible roles for combination therapy in CRAB infections; the results indicate that combination therapy could provide benefits in such cases [6,7,8,9,10,11,12]. In addition, two large trials compared the effects of treatment using colistin plus meropenem with treatment using colistin alone, among patients with severe CRAB infections. Both trials concluded that adding meropenem to colistin did not promote better clinical outcomes [7,8]. The IDSA panel favored the use of combination therapy for moderate-to-severe CRAB infections because of expected high bacterial burdens, the critical condition of patients, and the development of bacterial resistance. The combination of meropenem and colistin, i.e., without the addition of a third agent, was not suggested [13].

Sitafloxacin is a new-generation broad-spectrum oral fluoroquinolone. It has been approved by the Thai FDA and has been available for clinical use in Thailand since 2011. It has been shown to be very active against many Gram-positive, Gram-negative, and anaerobic clinical isolates, including strains resistant to other fluoroquinolones [14]. Sitafloxacin has shown good in vitro activity against MDR CRAB. It has also been shown to penetrate well into the epithelial lining fluid in patients who are critically ill with pneumonia [15].

For the present study, we designed a trial to compare the effectiveness of sitafloxacin–colistin–meropenem and colistin–meropenem combination therapies in patients with MDR CRAB infections. By doing so, we sought to assess the effectiveness of sitafloxacin-based combination therapy in the treatment of MDR CRAB, and thereby promote the development of a new treatment option against a resistant microorganism.

## 2. Objective

In this study, we sought to evaluate the efficacy and safety of sitafloxacin–colistin–meropenem, compared with that of colistin–meropenem, against CRAB HAP/VAP. The primary outcomes in this study were 7- and 14-day mortality rates; these were defined as numbers of patient deaths from causes which occurred within 7 or 14 days of the beginning of treatment. Secondary outcomes included the following: clinical cures lasting for at least 48 h; combinations of symptoms and signs (defervescence, decreasing FiO_2_, successful extubation or mechanical ventilator weaning); laboratory values (decreasing white blood cell count, procalcitonin, lactate); and organ function markers (Acute Physiology And Chronic Health Evaluation II (APACHE II) and the quick SOFA (qSOFA) score)). In addition to other secondary outcomes, the study considered microbiological clearance at 3 and 7 days as well as incidences of the following adverse drug effects: acute kidney injury (AKI); QT prolongation; acute diarrhea; *Clostridioides difficile* infection (CDI); and transaminitis.

The findings of this research may provide benefits for medical practitioners who may wish to apply an alternative treatment for CRAB HAP/VAP in healthcare settings in the future.

## 3. Results

Between April 2021 and April 2022, a total of 81 patients were enrolled. Four patients were excluded because they had additional bacterial infections (*n* = 2), bacteremia (*n* = 1), or a colistin drug allergy (*n* = 1). Of the 77 participants diagnosed with CRAB HAP/VAP who were then included in the trial, 40 were randomly assigned to combination treatment with sitafloxacin–colistin–meropenem; the remaining 37 received colistin–meropenem therapy. All of the participants were admitted to Rajavithi Hospital (Figure 1).

The demographics and baseline characteristics of all participants are presented in Table 1. The majority of the participants were men, with a mean age of 59.82 years (SD 17.11) and a mean BMI of 21.31 kg/m^2^ (SD 4.10). The majority of the participants had other medical illnesses (94.8%). The most common comorbidities were neurological disease (45.5%), followed by cardiovascular disease (35.1%), and malignancy (26%). There were no significant differences between the two treatment groups with respect to sex, age, or comorbidities other than neurological disorders.

With respect to disease severity, as indicated by APACHE II and qSOFA scores, mean scores of 12.95 ± 6.09 and 0.91 ± 0.69, respectively, were obtained. However, the mean APACHE-II score for the sitafloxacin–colistin–meropenem group was significantly higher than for the other group, indicating an increased mortality rate (14.33 ± 6.76 vs. 11.46 ± 4.95; *p* = 0.036). Significantly higher levels of procalcitonin and lactate were found in the first group (2.04 ± 3.72 vs. 0.35 ± 0.56, *p* = 0.007 and 1.97 ± 0.78 vs. 1.62 ± 0.55, *p* = 0.027, respectively). The white blood cell count was higher in the sitafloxacin–colistin–meropenem group, indicating a higher level of infection or inflammation (*p* < 0.001). In total, 59 participants (76.6%) with acute respiratory failure had received mechanical ventilation. Nine participants (11.7%) had a documented positive culture for CRAB in the sputum within the preceding 90 days. The incidence of previous carbapenem therapy (in the preceding 90 days) was 46.8%. The average duration of hospitalization before CRAB HAP/VAP was 12 days (IQR 8–26) (Table 1).

Table 2 shows the distribution of MIC values for meropenem and colistin among CRAB isolates. The majority of CRAB isolates (94.8%) showed MIC values higher than 16 µg/mL for meropenem, while 90.9% of CRAB isolates had colistin MIC values of ≤1 µg/mL. Concerning sitafloxacin susceptibility, sitafloxacin disk diffusion was not available in the Rajavithi hospital because of a shortage caused by the COVID-19 pandemic.

The primary outcome and the secondary outcomes of this study are presented in Table 3. The overall 7-day mortality was 7.5% in the intervention groups and 2.7% in the control group, with no significant difference (*p* = 0.616). In addition, with respect to 14-day mortality, there was no statistically significant difference between the two groups (10% vs. 10.8%, *p* = 1).

In terms of the secondary outcomes, we found that the majority of patients (58 of 77, 75.3%) had achieved clinical cure at the end of treatment. The incidence of clinical cure was significantly higher in the sitafloxacin–colistin–meropenem group compared with the colistin–meropenem group, as revealed by both intention-to-treat (87.5% vs. 62.2%; *p* = 0.016) and per-protocol analysis (87.2% vs. 67.7%; *p* = 0.049). Microbiological clearance was achieved by 40% and 47.5% of the intervention group at 3 days and 7 days, respectively; corresponding figures for the control group were 35.1% and 40.5%. These results therefore lacked statistical significance (*p* = 0.660 and *p* = 0.539, respectively).

We also evaluated the extent of clinical cures following initial antibiotic therapy after 3 and 7 days. Defervescence at 7 days in the sitafloxacin–colistin–meropenem group indicated decreases in body temperature which were significantly greater than in the colistin–meropenem group (37.32 ± 0.73 vs. 37.74 ± 1.08; *p* = 0.048). Among ventilated patients, weaning from mechanical ventilation was higher in the intervention group, but without any significant difference (45.2% vs. 22.2%; *p* = 0.067). The duration of colistin and meropenem was longer in the sitafloxacin–colistin–meropenem group.

The assessment of overall adverse events associated with the medication showed no significant differences between the two groups (62.5% vs. 67.6%; *p* = 0.642). However, nephrotoxicity was not significantly associated with combination therapy in the sitafloxacin-exposed group (35% vs. 51.4%; *p* = 0.147). Other adverse events in which no significant differences were found between the two groups included QT prolongation (20% vs. 24.3%; *p* = 0.648), acute diarrhea (12.5% vs. 18.9%; *p* = 0.438), *C. difficile* infection (*n* = 0), and transaminitis (35% vs. 32.4%; *p* = 0.812).

During the conduct of the clinical trial, seven participants were withdrawn for per-protocol analysis (intervention arm: 1; conventional arm: 6). The main reasons for treatment discontinuation were adverse events, mostly acute kidney injury (4/77; 5.2%), with one participant refusing dialysis treatment. Clinical deterioration led two participants (2.3%) to leave therapy prematurely to receive more optimal treatment. One recovering participant was transferred back to the referring hospital.

In the multiple logistic regression analysis presented in Table 4, no significant difference between the two groups is revealed with respect to mortality rate. However, significant differences with respect to clinical cures were revealed by intention-to-treat analysis (adjusted OR 3.93, 95% CI 1.21–12.78; *p* = 0.023) using multivariable analysis.

## 4. Materials and Methods

### 4.1. Ethics Considerations

This study was approved by the ethics committee on human research at Rajavithi Hospital (approval number: 63037) and was registered in the Thai clinical registry with the trial number TCTR20221221001.

### 4.2. Study Design

We conducted a randomized-control, two-arm, parallel-group, superiority, open-label study at Rajavithi Hospital, a tertiary-care hospital in Bangkok, Thailand, from 1 April 2021 to 30 April 2022. The study population consisted of hospitalized patients who were diagnosed with CRAB HAP/VAP.

### 4.3. Setting and Participants

Participants with CRAB HAP/VAP were identified by means of both clinical evaluation and assessment by a microbiology laboratory. Sputum samples were collected and sent to the lab for bacterial culture. *A. baumannii* was isolated from a direct colony using a matrix-assisted laser desorption/ionization time-of-flight (MALDI-TOF) mass spectrometer, enabling rapid identification.

For antimicrobial susceptibility tests, the broth microdilution method was used to determine the MIC according to the recommendation from The Clinical and Laboratory Standards Institute (CLSI). Carbapenem resistance was determined based on the CLSI interpretive criteria for meropenem nonsusceptibility as MIC ≥ 8 µg/mL, but *breakpoints* for *sitafloxacin* against *Acinetobacter* spp. were unavailable. Regarding sitafloxacin susceptibility testing, our attempts to obtain sitafloxacin standard powder and susceptibility test disks from the company were unsuccessful. The company rejected our request for sitafloxacin disk diffusion testing, citing the COVID-19 pandemic as the reason for their inability to fulfill the request.

Diagnosis of HAP/VAP was carried out according to the clinical practice guidelines issued in 2016 by the IDSA. Diagnosis was based on the presence of new lung infiltrate plus clinical evidence, the latter including new onset of fever, purulent sputum, leukocytosis, and decline in oxygenation [1].

To obtain samples acceptable for microbiological confirmation, expectorated sputum and endotracheal aspiration were used for culture and for the identification of bacteria. Samples were considered valid if they met the following quality criteria: polymorphonuclear counts of ≥25 and counts of squamous epithelial cells of <10 per field.

Inclusion criteria for the study were as follows: (1) hospitalized patients aged ≥18 years; (2) a clinical diagnosis of HAP/VAP, as defined by a new infiltrate identified using chest radiography, and associated with at least one of the following: purulent tracheal secretions, a temperature of 38.3 °C or higher, or white blood cell count higher than 10,000/µL; (3) for sputum or endotracheal aspiration samples with microbiological confirmation of *A. baumannii*, a cut-off point of ≥10^5^ colony/mL was considered [16].

Patients were excluded if they (1) had *A. baumannii* infection in organ sites in addition to pneumonia; (2) had bacteremic pneumonia; (3) had a sputum culture with colistin-resistant *A. baumannii* strains; (4) had co-existent acute kidney injury, chronic kidney disease at stage 3 or above, or liver failure; (5) were pregnant; or (6) had a history of colistin and fluoroquinolone allergy.

### 4.4. Randomization and Intervention

Enrolled participants were randomized with an allocation ratio of 1:1, then assigned by sequential order to one of two treatment groups: a control group and an intervention group. Participants in the control group received colistin–meropenem combination therapy; those in the intervention group received a combination of sitafloxacin plus colistin plus meropenem.

Dosages of antimicrobials were adjusted based on each patient’s renal function, expressed as creatinine clearance (CrCL) estimated using the Cockcroft–Gault equation. Patients with normal renal clearance received a 100 mg dose of sitafloxacin twice daily; those with CrCl of ≤50 mL/min received 50 mg twice daily; those with CrCl of 30–49 mL/min received 50 mg once daily; those with CrCl of 10–29 mL/min received a 50 mg dose every 48 h.

Patients with CrCl of ≥50 mL/min were prescribed 300 mg/day of colistin, divided into 2–3 doses. Patients with CrCl of 41–50 mL/min received a maintenance dose of either 150 mg every 12 h or 75–100 mg every 12 h; those with CrCl of 31–40 mL/min received a dose of 75–100 mg/day every 12 h; those with CrCl of 21–30 mL/min received a dose of 150 mg/day; those with CrCl of 11–20 mL/min received a dose of 100 mg/day every 24 h; those with CrCl of ≤10 mL/min received 75 mg of colistin every 24 h.

With respect to meropenem dosages, patients with CrCl of 51–90 mL/min received a standard dose of 1 g, administered every 8 h; those with CrCl of 26–50 mL/min received a 1 g dose every 12 h; those with CrCl of 10–25 mL/min received a dose of 0.5 g every 12 h; those with CrCl < 10 mL/min received 0.5 g every 24 h.

For the study population, demographic and clinical characteristic data were obtained that included age, gender, comorbidities, initial laboratory parameters, and previous experience of antimicrobial therapy. The severity of illness was measured using APACHE II and qSOFA score for sepsis.

In both the patient groups, the occurrence of the following adverse drug events was recorded: AKI, QT prolongation, acute diarrhea, *Clostridioides difficile* infection, and transaminitis. The incidence of adverse drug events in the two groups during treatment was then compared.

The duration of antimicrobial therapy was evaluated on a daily basis to ensure the shortest possible treatment. All participants in the per-protocol group were treated with a 7–14-day course of antimicrobial therapy. Treatment with colistin and meropenem could be terminated in the event of clinical cure or patient death.

Participants currently in clinical trials were also excluded if we later found that the inclusion criteria had not been met.

### 4.5. Follow-up and Definitions

The following data were recorded daily during the 14 days after treatment: vital signs, fractions of inspired oxygen, and the mechanical ventilation status of patients.

Laboratory values (white blood cell count, procalcitonin, lactate, AST, ALT, creatinine, QT interval on the electrocardiogram, sputum or endotracheal aspiration sample culture) and organ function markers (APACHE II score, qSOFA score) were determined on days 1, 3, and 7.

Responses to treatment were determined on days 3 and 7 after initiation of treatment by internists from the ward staff using clinical, biochemical, and microbiological parameters. Sputum samples were collected on days 1, 3, and 7 for assessment of microbiological clearance. Clinical parameters included symptoms and signs (defervescence, decreasing FiO_2_, successful extubation or mechanical ventilator weaning), laboratory values (decreasing white blood cell count, procalcitonin, lactate), and organ function markers (APACHE II score, qSOFA score). All data were collected on days 3 and 7.

The definition of clinical cure was the cessation of all antibiotics and survival for at least 48 h after the completion of antibiotics, without the need for resumption of antibiotic treatment [17].

Treatment failure was defined as the persistence and/or worsening of clinical signs or symptoms of pneumonia 72 h after treatment was initiated [18,19].

HAP was defined as pneumonia not incubating at the time of hospital admission and occurring 48 h or more after admission.

VAP was defined as pneumonia occurring >48 h after endotracheal intubation [20].

AKI was defined by any of the following criteria: (1) An increase in serum creatinine of 0.3 mg/dL or more within 48 h; (2) an increase in serum creatinine to 1.5 times the baseline or more, noted within the previous 7 days; or (3) a urine output of less than 0.5 mL/kg/h for a duration of 6 h [21].

The Acute Physiology and Chronic Health Evaluation II (APACHE II) is a widely used scoring system that measures the severity of disease for adult patients admitted to intensive care units (ICUs). The score ranges from 0 to 71, with higher scores indicating a more severe disease and a higher risk of death. With APACHE II scores of 0–15, 16–19, 20–30, and above 30, the possibility of mortality is 10, 15, 35, and 75 percent, respectively [22].

The qSOFA is a bedside prompt that assigns one point for each of the following three criteria: low blood pressure (SBP ≤ 100 mmHg); high respiratory rate (≥22 breaths per min); and altered mentation (Glasgow Coma Scale < 15). Total scores thus range between 0 and 3 points and are designed to identify patients with suspected infection who are at greater risk of a poor outcome outside the ICU. The authors of found that 24% of infected patients who scored 2 or 3 points on the qSOFA accounted for 70% of deaths [23].

### 4.6. Outcome Measure

The primary outcome was to evaluate the efficacy of the two treatments against CRAB HAP/VAP by considering all-cause mortality rates at 7 and 14 days in both the sitafloxacin–colistin–meropenem and colistin–meropenem treatment groups.

Secondary outcomes included microbiological clearance after 3 days and 7 days, the confirmation or absence of a clinical cure, and adverse events after treatment.

### 4.7. Statistic Analysis

Data analysis was performed using Stata software, version 14 (Stata Corp, College Station, TX, USA).

According to Katip W et al., 59.7% of HAP/VAP patients who received meropenem plus colistin experienced a clinical cure. We expected that, by adding sitafloxacin to this combination, we would achieve an incidence of clinical cure which was 30% higher. With 80% power and one-side 2.5% alpha, there were a total of 40 participants in each arm, allowing for a drop-out rate of 20%.

Categorical variables were expressed as counts and percentages. Continuous variables were demonstrated as mean and standard deviation or median and interquartile range. For comparison between variables, a chi-squared or Fisher’s exact test was used to compare categorical variables. For continuous variables, an independent *t*-test or a Mann–Whitney U test was used.

Variables were further entered into a logistic regression analysis to determine the efficacy of treatment in the sitafloxacin–colistin–meropenem and colistin–meropenem treatment groups, with both the primary outcome (7- and 14-day mortality) and the secondary outcomes presented as OR and 95% CI.

All tests of significance were two-tailed and considered significant when the *p*-value was expressed. Statistical significance was indicated by *p* < 0.05.

## 5. Discussion

In this study, we evaluated the efficacy of sitafloxacin–colistin–meropenem combination therapy compared with colistin–meropenem therapy in the treatment of CRAB HAP/VAP. This was the first study to use a randomized trial to assess the impact of adding sitafloxacin to a combination of colistin and meropenem for the treatment of MDR CRAB HAP/VAP.

Our results indicated that 7-day and 14-day mortality was not reduced by the addition of sitafloxacin to colistin–meropenem therapy. The all-cause mortality rate in the colistin–meropenem group in our trial was lower than that reported in previous studies [8,9,10,11]. This outcome might be explained by the younger ages of our participants and their lower APACHE II scores.

In terms of secondary outcomes, we found that the incidence of clinical cure was significantly higher in the sitafloxacin–colistin–meropenem group, compared with the colistin–meropenem group, both in intention-to-treat (87.5% vs. 62.2%; *p* = 0.016) and per-protocol analysis (87.2% vs. 67.7%; *p* = 0.049). Even though the clinical parameters in the first group at the beginning was worse than in the other groups, participants in the first group achieved more clinical cures. When we compared our results to those of previous studies, we found that the rate of clinical cure or treatment failure was the same as that reported for colistin–meropenem combination therapy (control group) [7,8,24]. The longer treatment duration in the sitafloxacin–colistin–meropenem group could be attributed to the patients in this group being more severely ill when compared to patients in the other group.

Defervescence was more prevalent in the sitafloxacin–colistin–meropenem group than in the colistin–meropenem group on day 7, but no difference was recorded on day 3. Because prolonged hospital stays are associated with infectious complications, defervescence also affects the time required to discharge hospitalized patients.

Therefore, according to the results of this trial, we may say that the addition of sitafloxacin to colistin–meropenem increases the clinical cure rate and reduces body temperature. These results may be due to the following reasons: first, even though CRAB is susceptible to colistin, serum concentrations of colistin may be variable and inadequate for bacterial activity in critically ill patients; second, antimicrobial resistance may emerge during therapy [13]. In cases of moderate-to-severe CRAB infections, treatment in combination with at least one other antimicrobial agent should therefore be considered to avoid treatment failure, according to supported current evidence [13].

In one study in Thailand in 2016, *A. baumannii* clinical samples were collected from in-patients at 13 tertiary-care hospitals to determine rates of susceptibilities. For sitafloxacin, against CRAB isolates, the MIC range was found to be ≤0.0625–8 µg/mL, with values for MIC_50_ and MIC_90_ of 1 µg/mL and 2 µg/mL, respectively. Almost all of the colistin-resistant isolates were susceptible to sitafloxacin (92.86%) [25].

Several studies have reported activity of sitafloxacin against *A. baumannii* [26,27,28]. In Thailand, researchers conducted an in vitro study and found that sitafloxacin was active against 58–88% of *A. baumannii* isolates, with a corresponding range of 18–21% for ciprofloxacin and levofloxacin [29]. Most combinations of sitafloxacin and rifampin, colistin, sulbactam, or tigecycline have been shown to exert synergistic and/or additive effects against CRAB. The authors showed that the rate of susceptibility for sitafloxacin was 91.67% against XDR-*A. baumannii* compared with corresponding rates for colistin, tigecycline, rifampin, and sulbactam of 62.5%, 54.17%, 41.67%, and 16.67%, respectively [30].

When considering the resistance of some strains of Gram-negative bacteria to sitafloxacin, the disk diffusion method may be used to determine sitafloxacin susceptibility. In Thailand, not all microbiology laboratories have the facilities to determine the MIC of sitafloxacin; this situation also applies at Rajavithi Hospital. In addition, the CLSI has not specified official inhibition zone diameters or MIC breakpoints for sitafloxacin against resistant Gram-negative bacilli. Because most laboratories are unable to determine the MIC of sitafloxacin, the disk diffusion method is used instead. It has been suggested that, in the case of resistant Gram-negative bacilli isolated from urine and blood, inhibition zone diameters of 16 mm or more and 18 mm or more, respectively, are susceptible to sitafloxacin [31]. In terms of the MIC breakpoint, more than one report has identified sitafloxacin at 2 µg/mL or less as susceptible to Gram-negative bacilli [26,31,32]. According to an annual drug susceptibility report from the Rajavithi Hospital’s microbiology department, in 2021, only 5% of *A.baumannii* was susceptible to meropenem, whereas 94% of *A.baumannii* was susceptible to colistin [33]. In addition, sitafloxacin susceptibility was stated to be 44% for *A. baumannii* in a 2018 report [34].

*A. baumannii* resistance mechanisms are complex, alone or in combination; they involve, inter alia, β-lactamases (classes A, B, and D), Amp C cephalosporinase, the modification of penicillin-binding proteins and porins, and the presence of efflux pumps [35,36,37,38]. Combination therapy creates therapeutic benefits because of potential synergistic mechanisms [35].

Our study conformed to the IDSA 2021 guidance that favored the use of combination therapy for moderate-to-severe CRAB infections for clinical benefit. Our results strongly support the idea of adding a third agent to colistin–meropenem, as part of the treatment regimen.

Although there was no difference between the two groups in terms of microbiological clearance, clinical cures were achieved in many cases without microbiological eradication. This might be explained as a consequence of reduced pathogenic organisms or inoculum reduction at the infection site, with continued colonization in the respiratory tract [39].

With respect to drug adverse effects, no significant differences between the groups were observed for all outcomes. Our study demonstrated that sitafloxacin might not increase the risk of AKI when added to colistin–meropenem treatment. Although a few participants (15.6%) had acute diarrhea, none had a positive stool test for CDI. In both groups, combination therapy did not increase the incidence of acute diarrhea, CDI, or transaminitis.

Though the sitafloxacin–colistin–meropenem combination achieved better rates of clinical cure compared with colistin–meropenem, and did so with a similar safety profile, any decision to adopt one regimen over the other should also consider other factors, such as antibiotic stewardship principles, as well as the cost and availability of drugs. The increased rates of clinical cure might indicate better efficacy, but more extensive studies with larger sample sizes are required to further validate these findings.

The principal strength of this study was that it was the first randomized control trial using sitafloxacin–colistin–meropenem combination therapy. In addition, we focused both on the specific disease and the pathogen, i.e., CRAB HAP/VAP, and we also evaluated clinical cures at both 3 and 7 days.

Our trial did involve several limitations. First, sample sizes became small after the dropout of participants, and this reduced the power of the study. Second, the current sitafloxacin MIC was not available in the Rajavithi laboratory, so we were unable to demonstrate the sitafloxacin susceptibility of *A.baumannii*. We recognize the importance of including MIC data to strengthen the overall validity and scope of our study. Third, our research work had a limited budget. For this reason, we conducted research using sitafloxacin for 7 days only. Finally, we found a problem relating to randomization. We randomly assigned two patients to the same group, next to each other, due to misnumbering. Future research should be encouraged to identify other appropriate treatment options for improving patient outcomes with respect to mortality and morbidity.

## 6. Conclusions

In this study, we found that combination therapy with sitafloxacin–colistin–meropenem did not demonstrate any superiority to colistin–meropenem treatment in terms of improved mortality rates in CRAB HAP/VAP patients, but it was associated with improved rates of clinical cure. No significant differences were found between the two kinds of treatment with respect to drug adverse events. These findings support the idea that treatment with sitafloxacin–colistin–meropenem may be chosen by medical practitioners to achieve better clinical outcomes.

## Figures and Tables

**Figure 1 antibiotics-13-00137-f001:**
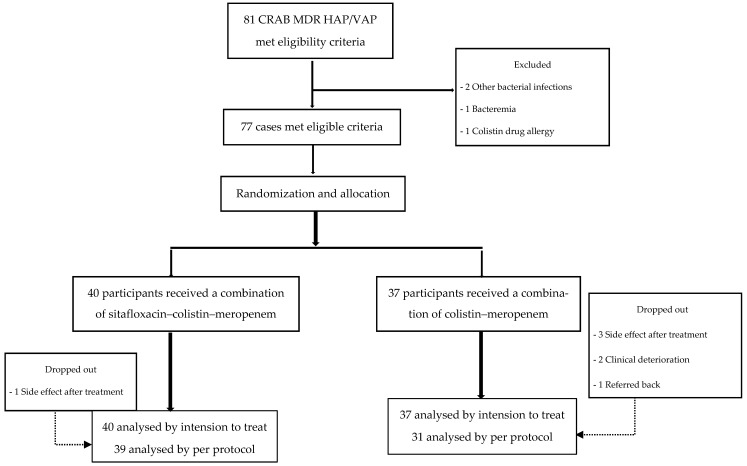
Enrollment and follow-up of the participants.

**Table 1 antibiotics-13-00137-t001:** Baseline characteristics of participants who received sitafloxacin–colistin–meropenem compared to colistin–meropenem combination therapy.

Characteristics	Total (*n* = 77)		Sitafloxacin–Colistin–Meropenem (*n* = 40)		Colistin–Meropenem (*n* = 37)		*p*-Value
Gender							
Male	56	(72.7)	27	(67.5)	29	(78.4)	0.248
Female	21	(27.3)	13	(32.5)	8	(21.6)	
Age (year)	59.82 ± 17.11		57.40 ± 18.26		62.43 ± 15.59		0.199
Bodyweight (kg)	56.83 ± 12.55		56.72 ± 10.39		56.95 ± 14.68		0.468
BMI (kg/m^2^)	21.31 ± 4.10		21.08 ± 3.47		21.56 ± 4.70		0.308
Underlying disease	73	(94.8)	38	(95.0)	35	(94.6)	1.000
Cardiovascular disease	27	(35.1)	13	(32.5)	14	(37.8)	0.624
Diabetes mellitus	13	(16.9)	8	(20.0)	5	(13.5)	0.448
Malignancy	20	(26.0)	8	(20.0)	12	(32.4)	0.214
Respiratory disease	6	(7.8)	4	(10.0)	2	(5.4)	0.676
Liver disease	4	(5.2)	2	(5.0)	2	(5.4)	1.000
Neurological disease	35	(45.5)	23	(57.5)	12	(32.4)	0.027
Others	6	(7.8)	2	(5)	4	(10.8)	0.419
Previous CRAB HAP/VAP	9	(11.7)	5	(12.5)	4	(10.8)	1.000
Previous carbapenem used	36	(46.8)	20	(50)	16	(43.2)	0.553
Mechanical ventilation	59	(76.6)	31	(77.5)	28	(75.7)	0.850
Duration of sitafloxacin (days)			8.75 ± 2.84				
Duration of colistin (days)	8.03 ± 2.80		8.75 ± 2.84		7.24 ± 2.58		0.009
Duration of meropenem (days)	8.03 ± 2.80		8.75 ± 2.84		7.24 ± 2.58		0.009
Hospitalization before HAP/VAP (days)	12	(8–26)	15	(9–36)	10	(8–19)	0.079
Body temperature (°C)	38.52 ± 0.96		38.69 ± 0.91		38.34 ± 0.98		0.115
FiO2	0.39 ± 0.13		0.39 ± 0.13		0.39 ± 0.13		0.840
APACHE II score	12.95 ± 6.09		14.33 ± 6.76		11.46 ± 4.95		0.036
qSOFA score	0.91 ± 0.69		1.05 ± 0.75		0.76 ± 0.6		0.063
White blood cell (cell/mm^3^)	14,124.55 ± 6007.66		16,675.25 ± 5877.33		11,367.03 ± 4871.2		<0.001
Procalcitonin (ng/mL)	1.23 ± 2.82		2.04 ± 3.72		0.35 ± 0.56		0.007
Lactate (mmol/L)	1.80 ± 0.70		1.97 ± 0.78		1.62 ± 0.55		0.027
Serum creatinine (mg/dL)	0.69 ± 0.25		0.65 ± 0.25		0.73 ± 0.24		0.151
GFR (mL/min/1.73 m^2^)	102.09 ± 23.37		107.55 ± 25.51		96.19 ± 19.49		0.031
CrCl (mL/min)	99.20 ± 45.53		111.41 ± 53.77		86.32 ± 30.57		0.015
AST (U/L)	59.81 ± 45.93		54.05 ± 37.85		66.03 ± 53.15		0.256
ALT (U/L)	54.97 ± 55.89		55.05 ± 47.24		54.89 ± 64.62		0.990
ALP (U/L)	137.45 ± 74.48		145.03 ± 87.09		129.27 ± 57.99		0.357
QT (msec)	451.96 ± 36.56		450.30 ± 36.36		453.76 ± 37.19		0.681

Data are presented as number (%), mean ± standard deviation, or median (interquartile range). *p*-value corresponds to the independent samples *t*-test, Mann–Whitney U test, chi-square test, or Fisher’s exact test.

**Table 2 antibiotics-13-00137-t002:** The MIC distribution of meropenem and colistin against CRAB isolates (*n* = 77).

Antimicrobial Agents	Total (*n* = 77)	Sitafloxacin–Colistin–Meropenem (*n* = 40)	Colistin–Meropenem (*n* = 37)
Meropenem			
>16 µg/mL	73 (94.8)	39 (97.5)	34 (91.9)
16 µg/mL	4 (5.2)	1 (2.5)	3 (8.1)
Colistin			
2 µg/mL	7 (9.1)	4 (10)	3 (8.1)
≤1 µg/mL	70 (90.9)	36 (90)	34 (91.9)

**Table 3 antibiotics-13-00137-t003:** Primary and secondary outcomes for patients who received sitafloxacin–colistin–meropenem, compared with those who received colistin–meropenem.

Outcome	Intention-to-Treat (ITT) Analysis							Per-Protocol (PP) Analysis						
	Total		Sitafloxacin–Colistin–Meropenem		Colistin–Meropenem		*p*-Value	Total		Sitafloxacin–Colistin–Meropenem		Colistin–Meropenem		*p*-Value
	(*n* = 77)		(*n* = 40)		(*n* = 37)			(*n* = 70)		(*n* = 39)		(*n* = 31)		
Primary outcome														
7-day mortality rate	4	(5.2)	3	(7.5)	1	(2.7)	0.616	4	(5.7)	3	(7.7)	1	(3.2)	0.624
14-day mortality rate	8	(10.4)	4	(10.0)	4	(10.0)	1.000	8	(11.4)	4	(10.3)	4	(12.9)	1.000
Secondary outcome														
Clinical cure	58	(75.3)	35	(87.5)	23	(62.2)	0.016	55	(78.6)	34	(87.2)	21	(67.7)	0.049
Microbiological clearance on day 3	29	(37.7)	16	(40.0)	13	(35.1)	0.660	26	(38.2)	16	(41)	10	(34.5)	0.583
Microbiological clearance on day 7	34	(44.2)	19	(47.5)	15	(40.5)	0.539	33	(53.2)	19	(52.8)	14	(53.8)	0.934
Drug adverse event	50	(64.9)	25	(62.5)	25	(67.6)	0.642	47	(69.1)	24	(63.2)	23	(76.7)	0.231
Nephrotoxicity	33	(42.9)	14	(35.0)	19	(51.4)	0.147	28	(40.6)	13	(33.3)	15	(50.0)	0.162
QT prolongation	17	(22.1)	8	(20.0)	9	(24.3)	0.648	17	(27.9)	8	(22.9)	9	(34.6)	0.311
Acute diarrhea	12	(15.6)	5	(12.5)	7	(18.9)	0.438	12	(18.8)	5	(13.5)	7	(25.9)	0.209
Elevation of liver enzyme from baseline	26	(33.8)	14	(35.0)	12	(32.4)	0.812	25	(39.7)	14	(38.9)	11	(40.7)	0.882
Clinical outcome														
Defervescence on day 3	37.81 ± 0.96		37.75 ± 0.84		37.87 ± 1.08		0.594	37.78 ± 0.83		37.69 ± 0.77		37.89 ± 0.90		0.318
Defervescence on day 7	37.52 ± 0.94		37.32 ± 0.73		37.74 ± 1.08		0.048	37.40 ± 0.67		37.23 ± 0.49		37.63 ± 0.80		0.032
FiO2 on day 3	0.37 ± 0.11		0.37 ± 0.09		0.38 ± 0.13		0.917	0.37 ± 0.08		0.37 ± 0.09		0.36 ± 0.08		0.559
FiO2 on day 7	0.36 ± 0.12		0.36 ± 0.10		0.36 ± 0.14		0.914	0.34 ± 0.09		0.35 ± 0.09		0.33 ± 0.09		0.522
Wean off mechanical ventilation (*n* = 58)	20	(34.5)	14	(45.2)	6	(22.2)	0.067	19	(35.2)	14	(45.2)	5	(21.7)	0.075
APACHE II score	11.55 ± 6.48		11.38 ± 6.25		11.73 ± 6.80		0.812	11.06 ± 6.37		11.03 ± 5.69		11.11 ± 7.26		0.961
qSOFA score	0.52 ± 0.66		0.43 ± 0.59		0.62 ± 0.72		0.194	0.41 ± 0.61		0.36 ± 0.54		0.46 ± 0.69		0.506
Laboratory on day 7														
White blood cell (cell/mm^3^)	9974.68 ± 4352.27		10,193.14 ± 4272.58		9691.48 ± 4519.07		0.656	9974.68 ± 4352.27		10,193.14 ± 4272.58		9691.48 ± 4519.07		0.656
Procalcitonin (ng/mL)	0.60 ± 1.95		0.28 ± 0.32		1.02 ± 2.93		0.204	0.39 ± 0.82		0.28 ± 0.32		0.54 ± 1.20		0.224
Lactate (mmol/L)	1.45 ± 0.54		1.44 ± 0.51		1.47 ± 0.58		0.872	1.45 ± 0.54		1.44 ± 0.51		1.47 ± 0.58		0.872
Serum creatinine (mg/dL)	0.96 ± 0.55		0.92 ± 0.54		1.00 ± 0.56		0.563	0.98 ± 0.60		0.93 ± 0.57		1.06 ± 0.63		0.385
Serum GFR (mL/min/1.73 m^2^)	86.78 ± 31.78		90.98 ± 34.05		82.24 ± 28.92		0.231	86.83 ± 34.25		91.5 ± 35.83		80.59 ± 31.59		0.214
CrCl (mL/min)	83.66 ± 58.16		94.74 ± 67.72		71.97 ± 43.95		0.088	86.27 ± 62.87		97.86 ± 70.49		71.26 ± 48.60		0.099
AST (U/L)	66.42 ± 65.28		59.80 ± 50.17		73.57 ± 78.54		0.359	66.65 ± 69.10		59.72 ± 52.23		75.89 ± 86.94		0.362
ALT (U/L)	48.12 ± 40.15		51.45 ± 34.64		44.42 ± 45.73		0.449	48.74 ± 40.57		53.86 ± 35.05		41.65 ± 46.97		0.246
ALP (U/L)	135.52 ± 83.89		139.85 ± 99.74		130.84 ± 63.53		0.641	140.03 ± 89.21		146.03 ± 102.46		132.04 ± 68.78		0.542
QT (msec)	455.86 ± 40.67		448.30 ± 32.39		464.03 ± 47.14		0.090	453.79 ± 40.53		448.77 ± 32.82		460.54 ± 48.94		0.266

Data are presented as number (%), mean ± standard deviation, or median (interquartile range). *p*-value corresponds to the Independent samples *t*-test, Mann–Whitney U test, chi-square test, or Fisher’s exact test.

**Table 4 antibiotics-13-00137-t004:** Multiple logistic regression analysis for primary and secondary outcomes for participants who received sitafloxacin-colistin-meropenem compared to colistin-meropenem combination therapy.

Outcome	Intention to Treat (ITT) Analysis						Per Protocol (PP) Analysis					
	Univariable Analysis			Multivariable Analysis			Univariable Analysis			Multivariable Analysis		
	OR	95% CI	*p*-Value	OR_adj_	95% CI	*p*-Value	OR	95% CI	*p*-Value	OR_adj_	95% CI	*p*-Value
Primary outcome												
7-day mortality rate	2.92	(0.29–29.38)	0.363	3.25	(0.31–33.60)	0.323	2.50	(0.25–25.3)	0.438	2.31	(0.16–34.3)	0.542
14-day mortality rate	0.29	(0.03–2.93)	0.294	0.27	(0.02–3.42)	0.312	0.77	(0.18–3.37)	0.730	0.75	(0.14–3.93)	0.730
Secondary outcome												
Clinical response	4.26	(1.35–13.44)	0.013	3.93	(1.21–12.78)	0.023	3.24	(0.97–10.79)	0.056	3.45	(0.92–12.97)	0.066
Microbiological clearance on day 3	1.23	(0.49–3.10)	0.660	1.41	(0.54–3.71)	0.487	1.32	(0.49–3.58)	0.583	1.43	(0.48–4.20)	0.521
Microbiological clearance on day 7	1.33	(0.54–3.27)	0.539	1.33	(0.52–3.42)	0.554	0.96	(0.35–2.63)	0.934	0.98	(0.34–2.83)	0.967
Drug adverse event	0.80	(0.31–2.05)	0.642	0.96	(0.35–2.64)	0.940	0.52	(0.18–1.53)	0.234	0.59	(0.18–1.94)	0.386
Nephrotoxicity	0.51	(0.20–1.27)	0.149	0.59	(0.22–1.59)	0.294	0.50	(0.19–1.33)	0.165	0.65	(0.22–1.91)	0.436
QT prolongation	0.78	(0.26–2.29)	0.648	0.93	(0.29–2.93)	0.896	0.56	(0.18–1.73)	0.314	0.65	(0.19–2.28)	0.505
Acute diarrhea	0.61	(0.18–2.13)	0.441	0.46	(0.12–1.81)	0.267	0.45	(0.13–1.60)	0.216	0.34	(0.08–1.41)	0.137
Elevation of liver enzyme from baseline	1.12	(0.44–2.89)	0.812	1.17	(0.43–3.17)	0.765	0.93	(0.33–2.56)	0.882	1.01	(0.34–3.04)	0.986

Abbreviations: OR, Odds Ratio; OR_adj_, Adjusted Odds Ratio; CI, confident interval; NA, data not applicable. Crude Odds Ratio estimated by Binary Logistic regression.

## Data Availability

Some of the data are available to be shared upon request.

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
