# Peer review of "(untitled)"

_antibiotics, 2024, doi:10.3390/antibiotics13020137_

Round 1
Reviewer 1 Report
Comments and Suggestions for Authors
The manuscript of Wantanatavatod & Wongkulab “Clinical efficiency ...” demands major changes for several reasons.
The authors write "There was no sitafloxacin MIC available in Rajavithi's lab, so we could not do this demonstrate the sensitivity of A. baumannii to sitafloxacin." This is a big limitation. Testing the drug without its in vitro evaluation?
1. Line 26 after Acinetobacter baumannii should be coma
2. Line 36 after Acinetobacter should be spp.
3. The sentence in Lines 41 to 43 needs rephrasing for clarity.
4. Line 45 what does OPD and IPD mean. Explain.
5. Line 67 in vitro should be written in italics, check in the whole text.
6. Line 72 only MIC, explanation is above.
7. Line 42,43, 73 use the same unit.
8. The sentence in Lines 41 to 43 needs rephrasing for clarity.
9. Line 118 since 2019 there is Clostridioides difficile not Clostridium.
10. There is no reference 42.
11. References: Revise for consistency. Lower case article titles; write correct microorganisms; italicize microorganisms.
Author Response
Thank you very much for taking the time to review this manuscript. Please find the detailed responses below and the corresponding revisions/corrections highlighted/in track changes in the re-submitted files.
Comments 1-11 : Thank you. I agree with these comments. Therefore, I have changed following your suggestions.
For sitafloxacin MIC, in our efforts to acquire sitafloxacin standard powder and susceptibility test disks for testing purposes, we encountered difficulties as the company declined our request. The reason provided for the rejection was the impact of the COVID-19 pandemic.
With Kind Regards,
Ms. Manasawee Wantanatavatod, MD
Reviewer 2 Report
Comments and Suggestions for Authors
This is a randomized controlled clinical trial whose primary aim was to evaluate the efficacy and safety of sitafloxacin with meropenem and colistin compared to meropenem and colistin alone in treating HAP/VAP due to CRAB infections.
The authors have included in the study a total number of 77 patients of which 40 were randomly assigned to sitafloxacin-colistin-meropenm group and 37 to colistin-meropenm group. The main results are the following: The addition of sitafloxacin did not improve 7-day and 14-day mortality, however, patients treated with sitafloxacin had higher rates of clinical cure and no differences were seen in terms of adverse events between the two groups.
The title clearly describes the study design and the paper, overall, is well written.
The abstract sums up the main contents of the work with coherence and effectiveness.
In the text, there is a summary of the past and current literature relevant to the aims of the study.
The methods section reports most of the information about the study. Results are presented. The discussion is fitting.
However, I would suggest a few pieces of information to be added/changed:
1. Objective: Please change leukocytes in white blood cells.
2. Methods: Which MIC criteria were used to evaluate the susceptibility profile of A. bauamnnii? Was it EUCAST or CLSI criteria? Please specify this information in the methods.
3. Statistical analysis: Lines 264-265: “Univariate and multivariate analyses were further entered into a logistic regression analysis to determine the efficacy of treatment in the sitafloxacin–colistin–meropenem and colistin–meropenem treatment groups, with both the primary outcome (7- and 14- day mortality) and the secondary outcomes presented as OR and 95% CI.” Please correct this sentence, variables are entered in the logistic regression not the univariate or multivariate analysis.
4. Results: Table 2, Please specify the day the laboratory values entered in the analysis belonged. The authors stated in the methods that lab values were taken on days 3 and 7 after treatment. The values used for Table 2 were the ones taken on the 7th day of treatment?
Author Response
Thank you very much for taking the time to review this manuscript. Please find the detailed responses below and the corresponding revisions/corrections highlighted/in track changes in the re-submitted files.
Comments 1: Thank you. I agree with these comments. Therefore, I have changed following your suggestions.
Comments 2: For antimicrobial susceptibility tests, broth microdilution method was used to determine the MIC according to the recommendation from The Clinical and Laboratory Standards Institute (CLSI). I haved added the detail.
Comments 3: Thank you. I agree with these comments. Therefore, I have changed following your suggestions.
Comments 4: I have incorporated Table 2 to depict the MIC distribution as suggested by reviewer 4. Table 2 has now been renumbered as Table 3. The laboratory values were obtained on the 7th day of treatment (Table 3). In the methodology section, I revised the text to state, 'All data were collected on days 1 and 7. Only the outcomes of microbiological clearance, defervescence, and FiO2 were gathered on days 1, 3, and 7."
With Kind Regards,
Ms. Manasawee Wantanatavatod, MD
Reviewer 3 Report
Comments and Suggestions for Authors
Dear Authors,
I have read the submitted paper paper titled " Clinical efficacy of sitafloxacin–colistin–meropenem and colistin–meropenem in patients with carbapenem-resistant and multidrug-resistant Acinetobacter baumannii hospital acquired pneumonia (HAP) / ventilator-associated pneumonia (VAP) in one super-tertiary hospital in Bangkok, Thailand: a randomized controlled trial".
The topic is of interest, methods are sound, results described well.
I have only minor comments.
All manuscript is rather long especially the Introduction
About the introduction some sentences should be described in the discussion i.e. line 63-69, 70-75, 76-84, 85-100 . Therefore I suggest you rewrite them in the appropriate section.
Author Response
Thank you very much for taking the time to review this manuscript. Please find the detailed responses below and the corresponding revisions/corrections highlighted/in track changes in the re-submitted files.
I have relocated the content originally found in lines 63-69, 70-75, 76-84, and 85-100 of the introduction to the discussion section.
With Kind Regards,
Ms. Manasawee Wantanatavatod, MD
Reviewer 4 Report
Comments and Suggestions for Authors
The authors themselves indicate the lack of in vitro tests that would indicate the minimum concentration of sitafloxacin to inhibit the study pathogens. I add that tests with in vitro associations are essential to indicate whether the association is positive (additive or synergistic) or negative (antagonistic). The clinic, given what is observed in patients, suggests that the associations must be confirmed in laboratory tests.

Author Response
Thank you very much for taking the time to review this manuscript. Please find the detailed responses below and the corresponding revisions/corrections highlighted/in track changes in the re-submitted files.
Comments 1-5: Thank you. I agree with these comments. Therefore, I have changed following your suggestions.
Comments 6: I have included Table 2 to illustrate the MIC distribution of meropenem and colistin.
Comment 7: For sitafloxacin MIC, in our efforts to acquire sitafloxacin standard powder and susceptibility test disks for testing purposes, we encountered difficulties as the company declined our request. The reason provided for the rejection was the impact of the COVID-19 pandemic.
Round 2
Reviewer 1 Report
Comments and Suggestions for Authors
I accept the current version of the manuscript. A major limitation of the study is the lack of evaluation of the MIC of sitafloxacin and this should be included as a limitation of the study.
Author Response
Thank you very much for taking the time to review this manuscript.
I have incorporated the limitation into the final paragraph of the discussion section, as you recommended.
With Kind Regards,
Ms. Manasawee Wantanatavatod, MD
Reviewer 4 Report
Comments and Suggestions for Authors
I believe the study has value for the clinic. All my questions were accepted.
Author Response
Thank you very much for taking the time to review this manuscript.
With Kind Regards,
Ms. Manasawee Wantanatavatod, MD